# HabITec: A Sociotechnical Space for Promoting the Application of Technology to Rehabilitation

**Elizabeth Kendall** [1,*], **Soo Oh** [1,2], **Delena Amsters** [1,2], **Mary Whitehead** [1,2], **Justin Hua** [1], **Paul Robinson** [1], **Dinesh Palipana** [1,3], **Andrew Gall** [1], **Ming Cheung** [4], **Leigh Ellen Potter** [5], **Derek Smith** [6] and **Brett Lightfoot** [7]

1. The Hopkins Centre, Menzies Health Institute Qld (MHIQ), Griffith University Nathan Campus, Nathan, QLD 4111, Australia; soo.oh@health.qld.gov.au (S.O.); delena.amsters@health.qld.gov.au (D.A.); mary.whitehead@health.qld.gov.au (M.W.); jhua8@live.com.au (J.H.); probinson85@live.com.au (P.R.); d.palipana@griffith.edu.au (D.P.); andrew.gall@griffithuni.edu.au (A.G.)
2. Metro South Health (Division of Rehabilitation, Dept of Occupational Therapy & Spinal Outreach Team), Princess Alexandra Hospital, Woolloongabba, QLD 4102, Australia
3. Griffith Centre for Biomedical and Rehabilitation Engineering (GCoRE), Griffith University, Gold Coast Campus, Parklands, QLD 4560, Australia
4. Griffith Centre for Design and Innovation Research (CDAIR), Griffith University, South Bank Campus, QLD 4101, Australia; m.cheung@griffith.edu.au
5. Innovative Design and Emerging Applications Lab, School of ICT, Griffith University, Nathan Campus, Nathan, QLD 4109, Australia; l.potter@griffith.edu.au
6. Advanced Design and Prototyping Technologies Institute, Griffith University, Gold Coast Campus, Parklands, QLD 4560, Australia; derek.smith@griffith.edu.au
7. National Director of Healthcare, Microsoft Australia, Brisbane, QLD 4000, Australia; brett.lightfoot@microsoft.com
* Correspondence: e.kendall@griffith.edu.au

**Abstract:** Society is currently facing unprecedented technological advances that simultaneously create opportunities and risks. Technology has the potential to revolutionize rehabilitation and redefine the way we think about disability. As more advanced technology becomes available, impairments and the environmental barriers that engender disability can be significantly mitigated. The opportunity to apply technology to rehabilitation following serious injuries or illnesses is becoming more evident. However, the translation of these innovations into practice remains limited and often inequitable. This situation is exacerbated by the fact that not all relevant parties are involved in the decision-making process. Our solution was to create a sociotechnical system, known as HabITec, where people with disabilities, practitioners, funders, researchers, designers and developers can work together and co-create new solutions. Sociotechnical thinking is collaborative, interdisciplinary, adaptive, problem-solving and focused on a shared set of goals. By applying a sociotechnical approach to the healthcare sector, we aimed to minimize the lag in translating new technologies into rehabilitation practice. This collaborative co-design process supports innovation and ensures that technological solutions are practical and meaningful, ethical, sustainable and contextualized. In this conceptual paper, we presented the HabITec model along with the empirical evidence and theories on which it has been built.

**Keywords:** assistive technology; rehabilitation; disability; independent living; sociotechnical design

## 1. Introduction

Technology is now fully integrated into our daily lives as a result of high-speed internet and the widespread commercialization of advanced products such as smartphones, networked appliances,

gaming consoles, artificial intelligence, and virtual assistants. Technology and the digital environment can connect people to each other regardless of their geographical location, cultural background, or other demographic differences. Despite the known presence of a 'digital divide' [1], Diamandis and Kotler [2] predicted that technology would bring abundance for all people and would allow us to solve the most complex problems in society. Indeed, people with all types of disabilities are now engaging with technology at unprecedented levels, making the application of technology to rehabilitation more feasible than ever before and offering vastly expanded opportunities to many people [3]. Some emerging technologies have the potential to radically transform rehabilitation through targeted, autonomous, and potentially inexpensive interventions [1].

When it works well, technology can be a seamless and integrated aspect of our lifestyle, woven into the fabric of society and capable of emancipating a large group of people who have been disabled by existing environments [4]. Access to technology is an important indicator of community integration, with Australian Bureau of Statistics[1] showing that more than 90% of Australians accessing the internet by a smartphone or laptop. Not surprisingly, people receiving rehabilitation are no less connected to technology. Indeed, technology has advanced to such a stage that recovery following traumatic injury or illness, rehabilitation, and independent living in the community could be revolutionized. However, barriers to uptake and sustainability prevent progress. Practitioners, people with disabilities, developers, and funders are willing to explore applications of technology, but they remain uncertain about how to make this happen in reality and have few guidelines to assist them. Ongoing dialogue between all stakeholders is essential to ensure that new technologies are developed in response to real rehabilitation challenges. Collaboration of this kind will ensure ideas are modified and refined until they are fit-for-purpose, the cost of implementation can be considered during the developmental process and necessary practice changes that will support implementation can be addressed.

The aim of this conceptual paper was to describe the development of a sociotechnical system that can address the challenges associated with applying technology in rehabilitation and maximize the likelihood that technology will deliver benefits to people with disabilities. We first reviewed the challenge of applying technology in rehabilitation across a range of contexts. We then described our local solution, HabITec, and showed how we have used sociotechnical spaces and processes to facilitate a system that can address these challenges. We then provided a typical example of how HabITec works and drew conclusions about the potential benefits associated with this approach.

## 2. The Technology Challenge in Rehabilitation

The World Health Organization[2] has predicted that, by 2030, over two billion people worldwide will require assistive technology, but only one in 10 currently have access to useful products that meet their needs. A recent study [5] found high technology usage rates amongst cardiac rehabilitation clients, with the majority (77%) being enthusiastic about receiving technology-aided rehabilitation. In another study [6], people in spinal cord injury rehabilitation universally valued technology because it made things easier, helped them to be organized, provided access to information and connections that would not otherwise be available and facilitated their participation in work and education, recreation and social events. Notwithstanding some negative impacts of technologies such as toxic social media, most believed that access to technology improved their mental health.

For people with intellectual and cognitive impairments, however, access to technology can be challenging. Indeed, research [1] has suggested that only about half of all people with intellectual disabilities access the internet or use a mobile phone, despite being able and willing to do so. Similarly, a study [7] of people with acquired brain injuries living in a shared supported accommodation found

---

[1]　See Australian Bureau of Statistics. https://www.abs.gov.au/ausstats/abs@.nsf/mf/8146.0. Accessed 28 October 2019.
[2]　See World Health Organisation Global Cooperation on Assistive Technology (GATE). https://www.who.int/disabilities/technology/gate/en/. Accessed 28 October 2019.

that they enjoyed using assistive devices and reported positive psychosocial benefits as a result of their technology use. The barriers to technology use among people with disability have been well-documented across a range of populations. In a recent systematic review [8] of assistive technology use by people with intellectual disability, 22 studies collectively identified 77 barriers and 56 facilitators. The most frequent barriers were associated with lack of funding and cost, lack of awareness about technology and inadequate assessment of need. A qualitative study of people with spinal cord injury [6] identified four major classes of barriers to the use of technology: (1) Limitations of the device/software in addressing the needs of the users; (2) personal limitations, such as impairments or circumstances, that restricted the use of the technology; (3) limitations in the information about technology and its use, including the ability to identify, acquire and understand technology; and (4) limitations created by the online environments, such as security, privacy and threats to safety. Similarly, barriers identified by people with acquired brain injuries [7] included inadequacies in the device and/or the user interface design, insufficient planning, consultation, support to use technology, and poor internet access. Barriers identified by people with strokes were predominantly associated with the difficulties of equipment setup [9]. Indeed, for many people with disabilities and their families, ease of setup and comfort of use were the most important factors in their decision about whether or not to use technology [10]. They were most likely to use technology if it was affordable and accessible, although younger people were more likely to be technology-tolerant and the barriers increased for older adults with disabilities [11].

Ironically, research has shown that the older population might be more willing to learn how to use technology, and in some instances, might be more prevalent users of technology than the younger generation [11]. The role of technology in the lives of the elderly is particularly important to acknowledge, as members of the older population are experiencing changes to their social circumstances with events such as retirement, death of a partner and perhaps physical limitations regarding their health. These factors can contribute to social isolation and loneliness which, in turn, can have a direct impact on physical and mental well-being. Technological advances, particularly the internet and social media, have created opportunities for the older adults to increase their connectivity and decrease social isolation.

Despite high levels of excitement about the use of technology in rehabilitation [12], actual knowledge about devices and technological solutions has been limited for not only the elderly, but also for most people with disabilities [6]. Self-efficacy has been posited as an important factor, as those with high levels of self-efficacy are more likely to engage in rehabilitation and to be early adopters of new technologies [13]. In most contexts, however, there is limited opportunity for either people with disabilities or practitioners to gain knowledge and confidence about technology. For instance, in the school setting, students with cerebral palsy, their parents, teachers and allied health professionals described major barriers to the use of assistive technology in the classroom [14]. They described the time required to establish technological solutions, delays in the acquisition of technology, limited access to support, lack of experience and awareness of technology and inadequate initial and ongoing training. There was also an absence of guidelines about the use of technology, limited funding opportunities and poor engagement of all stakeholders. More importantly, the impracticality and stigma associated with using some technologies in social settings, such as classrooms or lectures, are likely to present major barriers for students.

In the workplace setting, where technology-aided services might be expected to have the highest uptake, similar barriers have been found for employees using digital health promotion programs [15]. Workers reported adverse expectancies, fear of the internet, inadequate information and negative attitudes about the usefulness and usability of technology. The uptake of technology-based programs was low, even among high-risk employees who would benefit from the rapid delivery of e-interventions to address physical and mental health. The researchers concluded that translation of technological solutions into the workplace first required a collaborative approach that could promote awareness about technology, its efficacy and usability. The uptake of technology-mediated vocational rehabilitation by

injured workers was found to be hindered in the same way, resulting in calls for more evidence of effectiveness and efficiency, but also engagement from all stakeholders [16].

Not surprisingly, practitioners across all types of rehabilitation reported interest in the role of technology to facilitate rehabilitation and recovery, but their actual uptake of technological devices and methods in rehabilitation remains low [17]. There is a general awareness about the existence of highly advanced therapeutic technologies, such as exoskeletons, robotic rehabilitation devices, virtual reality and stimulation of the central or peripheral nervous systems [17]. However, the likelihood that these interventions will be successfully implemented into clinical practice, at least within the current macro-environment, is minimal.

Although there is widespread use of technology by rehabilitation practitioners in Australia, and even those working in rural areas [18], it is primarily used to support client contact, professional development and networking rather than as a core element of rehabilitation interventions. Researchers have documented a surprising lack of knowledge amongst practitioners about technology and how it might be used in rehabilitation, combined with little or no allocated time to learn [17] and minimal access to technology within the centres [11]. In the Netherlands, the implementation of "e-rehabilitation" has been plagued by lack of practitioner time, resources and knowledge, difficulty using technology and lack of perceived benefits, but also by organisational factors such as uncertainty about financial arrangements, privacy concerns and the organisation of care in ways that hinder technology use [19]. Other organisational barriers to the use of technology included interface problems and connectivity challenges and rigid procedures [10]. For most public rehabilitation centres or private practitioners, particularly in rural areas, the cost of purchasing, establishing and maintaining advanced technologies is well beyond their allocated budgets. Many centres have experienced additional organisational challenges such as a lack of infrastructure, particularly internet connectivity and limited availability of devices for purchase even if funds were identified [18].

Practitioners in stroke rehabilitation have reported that their use of technology would be positively influenced by strong evidence about its clinical effectiveness [11]. Indeed, research has shown that the opportunities generated by the technological revolution will not be realized in rehabilitation without investment in innovative research that can inform both future and current practice [4]. Although it is not unusual for practitioners to prescribe technology in the absence of any evidence [18], there is a paucity of empirical data to support technology-mediated interventions [1]. A recent review of studies about technology in spinal injury rehabilitation [17] found that most interventions involved specialized technologies that were rarely available in the hospital setting and could not be used by rehabilitation practitioners without access to technological expertise. Of the 73 studies, only five gathered data about the meaningfulness of the intervention to clients and practitioners, only two included an economic analysis that could guide implementation and none examined the workability of the technology within the rehabilitation setting.

## 3. HabITec: A Local Solution to Improve Uptake of Technology

HabITec is a joint initiative of Griffith University, Metro South Health, Microsoft Australia and several non-government organisations that support people with disability. The model emerged following a student practicum and research project with rehabilitation practitioners focused on the application of technology in rehabilitation [20]. Focus groups with therapists clearly identified a high level of frustration about their inability to identify and apply advanced technological solutions to the challenges they were observing in their clients. They reported many of the well-documented challenges, including lack of access, awareness, information, education and training about constantly evolving technology, inadequate funding schemes to support technology, inability to trial, store and update technology in a timely way within the hospital system, high abandonment rates, the need for peer support for end-users and the inability to customize solutions. HabITec itself was co-designed with people who have significant physical and/or cognitive/language impairments but are also experienced

in technology. The expert-user panel members draw on their own experiences of accessing health care, disability services and technologies to inform the way in which HabITec is delivered.

HabITec is both a physical space and a process that seeks to improve the uptake of technology in rehabilitation and community living with the aim of maximising recovery and capability. It is our local solution to the challenges associated with applying technology to the rehabilitation, participation and independence of people who have experienced changes in their physical, sensory and cognitive components as a result of brain injury, spinal cord injury, stroke or amputation. HabITec is based on the application of technology to two main settings (inpatient rehabilitation and post-discharge community). Within these settings, HabITec addresses three main aims: (1) Awareness and uptake of technology; (2) innovation in practice; and (3) bespoke solutions to improve recovery, independent living and participation in their occupation and within their environment. These aims, which are defined more fully in Table 1, allow HabITec to make existing technology more available to people with disabilities or find new ways of using existing technology, to modify and combine existing technologies in ways that enhance their utility and sustainability and to create a pathway for the development of new and innovative technology to address challenges. The technology available through HabITec may include simple interventions, such as trialling a stylus for limited hand function or implementing the accessibility features on a smart device through to applications of augmented reality, 3D printing, smart home systems and new software or hardware development. HabITec also focuses on the combination of multiple technologies to produce new solutions.

**Table 1.** Aims of the HabITec Sociotechnical System.

---

#### Aim 1: Growing Awareness and Uptake of Technology

- Exposure to the examples and information about the latest and available technology;
- A repository of interdisciplinary knowledge about new technologies, applications and usability;
- Identification of knowledge gaps about the cost, availability and effectiveness of technologies;
- Synthesis of existing evidence about the application of technology in disability and rehabilitation.

---

#### Aim 2: Promoting Innovation in Disability and Rehabilitation

- Surveillance of disruptive technologies that could advance recovery, rehabilitation, participation and independent living;
- Demonstrations of new technologies and cutting-edge advances that might influence the delivery or efficacy of rehabilitation or community initiatives;
- Engagement in the development of new technologies;
- Trials of new technology applied in rehabilitation or community settings.

---

#### Aim 3: Developing Bespoke Solutions

- Development of personalized applications of technology across a range of contexts;
- Engagement of practitioners, users, designers and developers with insurers and funders to identify appropriate and affordable solutions to identified challenges;
- Assessment of needs and identification of opportunities to apply technology to solving challenges;
- Promoting and fostering non-traditional professional alliances and collaborations to generate creative solutions.

---

HabITec is currently located in a hospital setting to ensure exposure to technological solutions as early as possible following entry to rehabilitation and to build awareness and capacity among rehabilitation practitioners. For people with acquired disabilities, rehabilitation is a place where people can safely trial and test different ways of doing things that were taken for granted before their injury or illness. It is important that HabITec is located in the hospital setting to maximize early exposure to technology. However, through our affiliated partnerships with non-government service providers and consumer organisations, it can also be made available in the future to people with disabilities already

living in the community. Indeed, queries and requests come to HabITec from a range of sources both inside and beyond the hospital environment.

Under the awareness banner, HabITec runs a number of programs designed to expose practitioners and people with disabilities to technology. These programs include demonstrations, information sessions, responding to direct queries, linking appropriate people, student-led placements, hack-a-thons and digital visioning workshops. To ensure adequate surveillance of innovation and its potential impact on rehabilitation or disability services, HabITec conducts regular systematic and scoping reviews of the latest technology in rehabilitation and manages a repository of relevant published studies. It will also host a platform where our ambassadors, people with disabilities and providers can share information and reviews about products and solutions. To promote innovation, HabITec hosts seminars from leading scientists who are working in fields such as brain-computer interfaces, extended reality and bionic devices. By supporting researchers to focus on the area of disability, HabITec aims to increase the amount and impact of responsible innovation.

In relation to the development of bespoke solutions, HabITec runs a problem-solving lab (The HabITec Lab). Engagement with this lab requires a stimulus challenge associated with the effectiveness of rehabilitation, participation in the community or independence in the home. The challenge could originate from any end-user (i.e., practitioner, person with disability, community advocate, policymaker/funder, service provider, developer/designer, family member). When a challenge is received, permission is first sought from the person with disability to focus on this challenge. The person with disability then becomes the "director" of the challenge. The challenge is defined, and the solution is carefully constructed using a collaborative and iterative design-thinking approach that involves ongoing input from all parties and continual review and refinement. The definition of these challenges is informed by the Person-Environment-Occupational (PEO) model [21]. 'Person' refers to the personal factors, values, experiences and the life stage. 'Environment' includes the physical, social, cultural and institutional environment in which people live, including their own home, community and society. 'Occupation' refers to the meaningful and functional tasks and activities required within people's work, leisure and social interactions. The model emphasizes the relationship between these three domains and the person's goals. It provides a framework to support people with disability and practitioners from multiple disciplines to develop a shared understanding of complexity. The PEO model relies on deep cooperation between practitioner and client, which fits well with the HabITec model of co-design.

## 4. A Sociotechnical Space to Support Collaboration

We embedded the PEO co-design process within a sociotechnical systems approach that has allowed us to view the challenges and potential solutions from multiple perspectives. It has also enabled HabITec to work across boundaries to find creative solutions. The sociotechnical approach facilitates not only a space where people could collaborate and share information, but also a process that simultaneously addresses ethical, social and technological challenges. Most importantly, this approach ensures the development of 'elegant' solutions (i.e., solutions that address barriers at multiple levels, reduce waste, deliver seamless coherent packages of technology and promote sustainability in context).

The notion of a sociotechnical system [22] was introduced by the Tavistock Institute of Human Relations in response to the challenges associated with introducing new technologies into coal mining in the UK during the 1950s. For nearly 70 years, sociotechnical systems thinking has been applied to a range of areas where new technology has been introduced to a workplace. It has even been used more broadly across many areas of design. This approach seemed to be the most appropriate overarching framework for HabITec because it acknowledges the presence of two independent but related systems that both influence uptake of new solutions, sometimes in contradictory ways. There are five key characteristics of a sociotechnical system, namely (1) clearly defined internal and external environments, (2) interdependent components, (3) an adaptive goal-oriented focus, (4) choices about

how to reach those goals and (5) a desire to optimize success through cooperation. Davis and his colleagues [23] described a sociotechnical system as a set of goals and associated metrics, a cluster of people with varying attitudes and skills, a range of technologies and tools, a physical environment, relevant cultural assumptions and a defined set of processes and practices. They acknowledged that this system is likely to be located within a wider context that usually includes a regulatory framework, broader stakeholders and financial constraints. This conceptualisation mirrored our circumstances and our objective in developing HabITec. Our first goal in addressing the barriers to technology uptake was simple awareness and exposure for all relevant stakeholders. Our second goal was to build collaboration across the boundaries that separated these different stakeholders. In determining the stakeholders who needed to be involved in HabITec, we focused on four types of "assistive technology professionals", namely (1) health practitioners who assess, prescribe and use technology in rehabilitation; (2) advocates and community providers who assist with the implementation and maintenance of technology in context; (3) designers, developers, engineers and researchers who contribute to innovation, technological advances and opportunities and (4) funders and suppliers of technology who resource and regulate the use of technology [24].

We recognized that the boundaries between these professionals were created by different approaches, drivers and philosophical beliefs, and that HabITec should provide a place where different professional cultures and orientations could be named, shared and explored to generate cooperation. Our aim was to create a shared culture focused on the central stakeholders (i.e., people with disability). Through our partnerships with technology experts such as Microsoft Australia, we were able to explore a range of different technologies and tools with which other stakeholders were not currently familiar and were unlikely to come across in everyday practice. We were also able to explore the limits and strengths of those technologies within a physical space that was highly visible and accessible to both practitioners and people with disabilities and suitable for trialling and testing without risk or cost.

## 5. Sociotechnical Processes: Frameworks for Design

When all these professionals work in unison with people who have a disability at the centre of the process, it is likely that assistive technology may become truly emancipatory. Thus, once our sociotechnical space was created, we needed to identify frameworks to guide the way in which we worked within that space. A common language and overarching framework were needed to allow shared understanding across multiple stakeholders, the production of meaningful technology and smooth implementation into rehabilitation practice and policy [23]. Our HabITec Lab was based on the documented need for a "transaction space for the exchange of knowledge" [25] and a conceptual framework that systematically explores the relationships between end-users (e.g., characteristics, needs, motivations), activities (e.g., daily living, work, leisure), and technologies (e.g., equipment, applications/software, connectivity) [26]. The process melded collaborative inclusive person-centred planning and solution-focused thinking with the notion of universal design. We acknowledged that the development of suitable and sustainable assistive solutions is not an isolated event but must be an iterative collaborative person-centred process during which creative solutions are trialled and refined over time with input from all stakeholders. This collaborative user-centred model is in line with Australia's National Safety and Quality Health Service Standards where the person is at the centre of all decision-making, allowing solutions to be tailored to meet their specific needs. Importantly, this approach enhances the sustainability, transferability and practicality of any solutions. In the technology field, an important notion is that of "zero-effort" technologies [24], meaning technologies that are unobtrusive, non-stigmatising and cause minimal disruption to the daily lives of those who use them. When designed using person-centred principles, technology is more likely to be applied and sustained with ease.

Through our partnership with Microsoft Australia, we adopted a "hacker" mindset of "design for one and build for many". "Hacking" comes with many negative connotations, but its origins are in

finding ways to improve systems. Hackers have the ability to see all possible futures, to recognize complex signs and patterns, to look at situations from every perspective and to explore all possibilities. They constantly seek ways to improve and future-proof systems. The value of the hacking mindset has been harnessed in fields such as cybersecurity and law enforcement, but also in business development, where courses are now being delivered to assist people in adopting this approach. By combining the hacking approach with the translation of new concepts into future designs that can benefit many other people, our approach also contributes to the overarching principles of universal design as required by the United Nations Convention on the Rights of People with Disability (2006).

Universal design is an important concept in the disability field and is based on well-established principles and practices. This approach focuses on ensuring that all products, places and services, including technology, are developed so most people can use them with minimal adaptation or modification [3]. Although it applies across multiple sectors, universal design is most often used in relation to the design of built environments that afford access to people of all abilities across a wide range of situations with minimal need for adaptation. However, in this era of rapid technological growth, people with disability are often faced with an expanding range of products that can result in confusion and inertia. In reality, assistive technology is likely to be more than any single product or device. It is more likely to be a system where a number of devices, people and environments interact to form a solution that promotes recovery or independence. Successful solutions are likely to integrate a range of specialist, bespoke and mainstream products [9] and, once combined, overall accessibility requirements may change.

A useful example to demonstrate the HabITec process is the application of smart home technology. Connectivity is a mainstream technology that can be used in any household, but if designed with a specific user in mind, it can be life-changing for people with mobility or cognitive impairments. However, in the case of disability, solutions must be integrated and focused on the context and activities of the person at the centre of the process. In engaging stakeholders in the design of a smart environment, HabITec teams systematically evaluate the ease of access, acceptability and cost efficiencies of the solution. By collaborating with hardware engineers, software designers and funders, practitioners and people with disabilities can develop bespoke individualized solutions, but also new knowledge about how to integrate technologies for better universally available devices. Over time, our focus on "personalized hacking" can create a repository of knowledge to guide the design of future technology, thus, contributing to broader social justice.

The HabITec principles can be challenging to maintain in the face of such an extensive and rapidly changing range of technologies that reach the marketplace through diverse developmental and commercial pathways, as well as in the context of severe and varied levels of impairment. To complicate this picture, technologies are often quickly superseded or refined and often represent only a small portion of a rehabilitation solution.

Without a solid structure and framework to guide decision-making, assistive technology can become susceptible to the dominance of "dazzle" over clinical evidence. We are acutely aware of the impact of "dazzle" in HabITec. As technology enters new phases of developmental maturity and exploration of its boundaries, the potential for dazzle will increase in the disability sector and will prompt an increasing need to attend to the ethical implications of technological solutions. Some researchers [27] have argued for a proactive ethical framework based on four major qualities: (1) Minimization of power imbalances, (2) compliance with biomedical ethical frameworks, (3) translational capacity and (4) awareness of broader social forces and requirements. Others have pointed out the importance of monitoring negative social messages where disability is promoted as a deviance that can be ameliorated by the positive power of technology [28]. Our approach draws on both microethics (i.e., the ethics of trustworthiness) and macroethics (i.e., the ethics of social justice) [29] to address the challenges of power, responsible development, sustainability and translation and social implications.

At the micro-level, trustworthiness comes from attention to privacy, safety, robustness, security and protection of end-users. In terms of ensuring trustworthiness (or dependability), the Sommerville

and Dewsbury framework [30] is helpful and functional. This framework takes a sociotechnical view of dependability that is not just about the hardware and software, but also considers a reflection of how well the solution fits into the environment and works for the particular individual in their context. In designing a technical solution for a person with disability, this framework recommends continual analysis against four key attributes—fitness-for-purpose, trust, acceptability and adaptability—each with multiple sub-attributes. Sommerville and Dewsbury provided a set of prompt questions and guidelines that can drive the design process and assist in selecting the attributes that are of highest priority. This process guides our decision-making in HabITec through a series of worksheets designed to stimulate relevant questions.

In terms of the broader ethics of social justice, it is important to pay attention to equity, access, impact, fairness and social responsibility. The European Union recently adopted a Responsible Research and Innovation Manifesto, the key premise of which is that science (and technology) is the basis for a better future, but it must be carefully directed in accordance with societal values and involving as many stakeholders as possible [31]. This approach requires a coordinated interdisciplinary action based on multiple perspectives that can appeal to and engage many different stakeholders. It also needs to renegotiate the boundaries that separate different sectors and form new alliances to support healthy innovation. Most importantly, it needs to be sensitive to the experience of disability and the way in which individuals make sense of their circumstances. Finally, it needs to provide a realistic analysis of the promissory visions of technology advocates to confirm they are grounded in the reality of life with a disability. Being based in a collaboration between the public health system and the resource-poor non-government sector, we also recognized the inherent tension between accountability for public expenditure and the quest for technological solutions. Thus, we added cost efficiency as an important performance indicator of our solutions. Each of these broad social challenges is listed in a HabITec checklist and forms the basis of our decision-making before solutions are finalized.

## 6. A Typical Example

To illustrate the way in which HabITec operates using this structure and process, a simple example has been described below. When an individual approaches or is referred to HabITec, his or her goals are explored with the occupational therapist using the Person Environment Occupation (PEO) framework. For example, in the case of a person who has slower cognitive processing speed, visual deficits and poor hand function following a neuromuscular condition, the goal might be to stay connected with society and friends and attend to personal tasks on the internet. The PEO framework will provide a full analysis of how that person operates in his or her environment to achieve those specific tasks. Once that information is available, the person becomes the director of the challenge, hosting supported meetings with key stakeholders from each area. In the initial phases, researchers are engaged to conduct rapid reviews of the literature searching for solutions. Developers and providers are engaged to provide advice on existing products and devices that may address the goals. Designers are engaged to identify new possibilities. Funders are contacted to examine the capacity to purchase solutions. A range of other practitioners and community providers might be engaged as needed, along with family and friends as determined by the person. Students may also be engaged if relevant.

After initial briefings and engagements have occurred, the hacking process begins. A digital visioning workshop will be convened at the most relevant location and with all relevant parties present. The challenge will be described and demonstrated, and potential solutions will be explored. Homework will be assigned, and a follow-up meeting will be organized to further develop, trial or select a solution. In designing a successful solution, the Sommerville and Dewsbury framework prompts us to examine the fitness-for-purpose, trust, acceptability and adaptability of the solution. If voice control and a virtual assistant are chosen, for example, the framework will force us to test this set of solution in different rooms and contexts. It will examine the level of privacy required and the ability of the technology to meet its purpose across a range of uses (i.e., sending emails, playing music). The acceptability and durability of different technologies will be examined for the user within his or

her social norms and expectations. Finally, the adaptability and maintenance of the solution will be explored across different circumstances and over time.

During this process, broader social implications will be examined (e.g., stigma, equity, cost, potential consequences for the social network, etc.). With permission of the person, our consumer panel will examine the solution in terms of broader usability and portability. Discussions will then be conducted with researchers, designers and developers about how to improve the solution or trigger new innovations in this area, as well as ways of making it accessible for future users. The eventual solution (and any evidence of effectiveness) will be shared with the online community, practitioners and funders. The person at the centre of this process may choose to become an ambassador to promote future consumer-demand for technological solutions.

The further development of any solutions may involve more complicated processes, such as intellectual property assignment, patent registration, approval of therapeutic goods and business proposals. HabITec has not yet addressed any issues of this nature, but it is recognized that a pathway is needed. We are currently developing this pathway but have noted the importance of applying the same principles to avoid power imbalances, stigmatising processes, inequitable benefits and unexpected consequences. HabITec is actively seeking a solution where ownership of innovation is equitably distributed in accordance with a collaborative co-design framework.

## 7. Benefits and Next Steps

Over the last half century, technology has become an integral part of our daily lives and offers the potential to change the way disability is understood and experienced. However, the design of technological solutions often lacks engagement of all relevant stakeholders, particularly people with disabilities. In this conceptual paper, we described HabITec, our local sociotechnical system that provides a structured opportunity for collaboration between developers, researchers, end-users, clinicians and policy stakeholders. HabITec aims to maximize the benefits of technology for both individuals with disability and society as a whole. Initial work with practitioners and rehabilitation clients confirmed the need for a vehicle such as HabITec to capitalize on major technological advances. In particular, HabITec responds to the urgent need to produce a more "tech-savvy" workforce in rehabilitation, something that has been noted by key researchers in this field [7] and the current workforce at the Princess Alexandra Hospital. HabITec provides a vehicle for this education and awareness of available technologies. It also addresses the systemic challenges that prevent uptake of technology in rehabilitation by building trust and acceptance in technology and evidence about effectiveness to underpin investment.

At a systemic level, HabITec can also contribute to the process of breaking down artificial boundaries between inpatient rehabilitation programs and the longer-term personal decisions made by people with disabilities as they negotiate their environments. There is a tenuous relationship between rehabilitation and disability services, with the former having historically grown out of a medical approach to treatment and the latter being driven by a consumer-driven social emancipation model. This gap between rehabilitation and disability services can create an artificial blockage for the rehabilitation journey. Technological solutions developed during the rehabilitation process need to be maintained and managed in the community, but community providers are often not engaged in the development and design process.

To maintain a consumer focus in the face of excitement about new innovations, HabITec is led and developed by a panel of people with disability and follows a person-centred planning and design process. More importantly, HabITec offers an opportunity to generate a demand-driven system where people with disabilities are placed at the centre of the design process and are also driving broader change and creating a marketplace for improved technology. The opportunity to engage closely with end-users is rare and highly valuable for developers and policymakers. For people with disability and practitioners, HabITec is an opportunity to invest heavily in those developers who genuinely want to see useful and meaningful application of their devices to the improvement of people's lives.

We are currently collecting data that will allow us to track the impact of HabITec and the change in both practitioner and consumer-driven demand for technology in the rehabilitation setting. Our future research aims to explore the experience of all stakeholders in the process of coming together to close the technological gap in rehabilitation and disability services. Our early observations suggest that these multifaceted benefits are made possible by our commitment to a sociotechnical systems approach. This approach has allowed us to create both a space and a process that achieves our vision of improving access to responsible technologies, building a "tech-savvy" rehabilitation workforce and providing tailored bespoke solutions to assist people with a disability to reach their rehabilitation goals, participate in society and improve their quality of life.

**Author Contributions:** All authors have contributed to the conceptualization and reviewing of this paper. E.K. took primary responsibility for drafting the paper with S.O. and D.A. All authors play an active role in the development and delivery of HabITec. E.K., S.O. and M.W. developed the concept of HabITec. J.H., P.R., D.P. and A.G. are current Ambassadors driving the delivery of HabITec. M.C., L.E.P., D.S. and B.L. are critical partners in design, development and innovation.

**Funding:** HabITec is an initiative of The Hopkins Centre, with funding, support and engagement from the Motor Accident Insurance Commission, Metro South Health, Griffith University, and Microsoft Australia.

**Conflicts of Interest:** The authors declare no conflict of interest.

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
