# Peer review of "HabITec: A Sociotechnical Space for Promoting the Application of Technology to Rehabilitation"

_societies, doi:10.3390/soc9040074_

Round 1
Reviewer 1 Report
Thank you to the editors of Societies for the opportunity to read this interesting and important manuscript. The authors and their colleagues are the stewards of a critically important program for rehabilitation. The work they describe has the potential to be transformative and to unblock persistent barriers to bridging technological, social, clinical and implementation divides. There is an enormous need for papers like this one. The complex and changing funding environment for disability, rehabilitation and assistive technology worldwide and Australia specifically require new evidence and practice-ready guidance about how to make sense of complex interplay between what people need, what they can get, how it's provided - and how that all comes together in implementation. Accordingly, I was excited to read such a manuscript and I hope it finds its way to publication in a revised form.
I structure my feedback below to correspond to the criteria provided to be as reviewer. Some general recommendations for alternative structures are presented, along with a line by line summary of minor considerations that will further strengthen the manuscript.
I was guided by the following description of Societies' expectations for 'articles':
Articles: Original research manuscripts. The journal considers all original research manuscripts provided that the work reports scientifically sound experiments and provides a substantial amount of new information. Authors should not unnecessarily divide their work into several related manuscripts, although Short Communications of preliminary, but significant, results will be considered. Quality and impact of the study will be considered during peer review.
Introduction.
The introduction explores several broad themes: the pervasiveness of technology in society in general, its potential, its use in rehabilitation, technology in cogitative and intellectual disability, technology and ageing. Then it turns to barriers to implementation, highlighting CP, self-efficacy. Another paragraph explores schools, then workplaces. Then you present implementation challenges in rehabilitation. The penultimate paragraph highlights how evidence should but doesn't inform practice, and examines the scenario for stroke and SCI. Later, section 3 (sic 4) extends the theoretical basis in the body of the paper and is contrasted with commentary on the application of existing frameworks to the current model.
The introduction makes for interesting reading, and introduces many concepts and themes that describe complexity in application of technology in disability and rehabilitation care. However, the problems are introduced very broadly, without offering much in the way of new information about organizing knowledge about the issues or what we might do about them. This illustrates my overall concern with the paper: I can't tell what it was you wanted to do with the manuscript as an article.
Based on the guidelines for authors, an article should report original research and 'sound experiments'. I believe the overall HabITec model could be described as such. Accordingly, I have no doubt that some of the work described in this paper - and the ongoing work in your lab - satisfies the criteria of being scientifically sound. I am sure that new information would arise by looking at the overall HabITec model from its theoretical underpinnings, to its implementation, and to its outcomes. However, as it is presented, I am concerned that it does not meet the requirements for Societies.
Methods:
Part 2 describes the HabITec lab. Description of partners is repeated on lines 148 and 181. Overall I find this description of HabITec lab to be clear and useful.
Results:
Part 3/4 Socio-technical processes: frameworks for design builds on what's presented in part 1, mixing descriptions of theoretical frameworks from literature and practice with descriptions of HabITech. line 293 for example introduces the concept of user-centric, collaborative, ongoing processes as an imperative, but it's not clear whether this is a finding from either a) the literature or b) your work - or c) taken as axiomatic.
Conclusion
4/5 conclusion - Overall, the conclusions either don't clearly arise from what you present as findings (either from the literature or your own work). Whatever the structure of the paper, that is an essential criteria. It presents findings that are either not mentioned or unclear from the body of the paper, which should be incorporated either into the conceptual basis, or findings, or whatever makes sense in a new structure. The conclusions should summarise either the concept presented, or the findings - or potentially both, depending on the revised structure.
Abstract
The abstract invokes use of 'we' and 'us' in a very general sense. What is meant by 'we' here? Later, you argue that some people are excluded, implying there is a marginalised subset of 'us'. Please revise. What is meant by territory, here? I suggest simplifying this: More technology creates opportunities and risks, for example.
Line 10 - grammatically, you're saying 'environment can be reduced'. Consider Advanced technology can mitigate environmental barriers and impairments that might otherwise result in disability, or similar. Consider whether you have introduced the concept of disability sufficiently to make this case yet. Would a line like 'disability arises from the relationship between impaired function and environments that are inaccessible' help, for example?
Line 13 - Grammatically, you've argued that people with disabilities (sic) are an industry. Please reconsider. Would a simpler sentence suffice here? Anyway I think these 2 sentences say the same thing: Available technology is not available to all, largely due to implementation barriers, and people miss out.
Line 15 - what conversations?
Line 16 - use a sentence to briefly describe socio-technical spaces/theories here.
There's a conceptual mix of disability and rehabilitation here that needs careful consideration. Is HabITec about one, the other, or both?
General remarks.
Taking Societies' recommended structure as a starting point, I have offered some thoughts above on the paper overall and its current suitability for publication as an article.
Taking the abstract, I was expecting the aim to be exploring: address(ing) the lag in translating new technologies into rehabilitation practice, with a presentation on results that describe how the HabITec model achieved that goal, and some interpretation of findings in light of the comprehensive review presented. Later, aims for the lab overall, not the paper, are described. More general aims and theoretical bases for your practice appear throughout, especially in the last 3 sections.
Put simply, the structure didn't follow what would normally be expected, and was therefore difficult to understand as a cohesive article. I strongly suggest the authors consider restructuring the paper to present clearer aims, a logical method, and results that are specific enough to be verifiable on the basis of what you've done, but general enough to be translatable to others interested in implementation challenges in technology and rehabilitation. Another option is to consider re submission as a conceptual paper, using the instructions for authors either way.
If you persist with a research article, I would like to see a more classical presentation of your work. After a simplified introduction based around logical subheadings and potentially a tabular presentation of the overall taxonomy of your logic, present an aim. The aim could either be to describe the model and present a generalizable set of theories about how it might work and for whom OR to describe the habitech model and present specific results and interpret them in context.
Method: either present a method about how you've set about to understand how the approach works to present findings or the overall HabITec approach as the method.
Results: present specific or illustrative results arising from the intervention, or potentially, present the overall model as a result, if the method describes how you determined what the HabITec model should be.
I reiterate my introductory comments that I find the HabITec model to be exciting, timely, and even transformative for rehabilitation practice. Accordingly, I hope the authors pursue a resubmission and that the editors afford the opportunity to do so.
I extend my regards to the authors of the manuscripts and thank them for their important work and contributions to the many fields for whom this paper will be interesting.
Author Response
We have reviewed the manuscript to address the helpful comments of this reviewer. We appreciate the effort taken by the reviewer to examine our manuscript in depth.
The reviewer's primary comment is about whether or not the manuscript should be an empirical or conceptual paper. We have elected to submit the article as a conceptual paper purely because it is too early in the process of HabITec to have collected reliable and robust evidence. This is underway at present with HREC clearance, but would not be useful yet. We acknowledge the reviewers comment and hope to produce an empirical paper soon.
To address the comments of the reviewer we have attended to the following specific issues:
I can't tell what it was you wanted to do with the manuscript as an article.
We have strengthened the aims of the manuscript separate to the aims of HabITec. We have clarified the purpose of the manuscript in the abstract and introduction.
it's not clear whether this is a finding from either a) the literature or b) your work - or c) taken as axiomatic.
We have clarified that the frameworks in the paper are from literature but also guide our practice and we have demonstrated how using an example.
The conclusions should summarise either the concept presented
We have provided an example prior to the conclusions and have changed the heading of Conclusions to Benefits and Conclusions. We have stated what we have included in the paper and we have related the benefits more clearly back to the original problem.
Grammatical errors in Abstract
We have corrected the points made by the reviewer.
There's a conceptual mix of disability and rehabilitation here that needs careful consideration.
We have explicitly stated that the intersection of disability and rehabilitation is important and that the aim of HabITec is to span this artificial boundary.
Reviewer 2 Report
General Structure:
Well written and concise Intro is nicely setup Clear description of stakeholders in the program The goals of program are clear The relationship between program and Microsoft Australia is direct. I enjoyed your overview of the socio-technical system.
Areas to Consider Improving
The description of HabITec is shown twice (lines 148 + 181). I recommend streamlining this a little more. Consider starting with the problem state under the HabITec section. What needs surfaced that led to the development of HabITec? What was observed and how did you plan to tackle this? Then dive more deeply into the model itself. This offers a stronger narrative to the importance of the program. Also, does HabITec support all disability types? Are there any limitations to what technology you can support? I enjoyed reading about the "hacker" mindset of "designing for one and build for many." I would like you to expand on this more for three reasons. First, make clear how this does (or does not) fits into universal design and zero-effort technologies. Second, build a stronger connection between this statement and the rest of your paragraph. The paragraph feels like it abruptly switches from what you are building to what other researchers have observed in the past. Consider restructuring the flow to make this less abrupt. Otherwise, your messaging becomes lost in the read. Third, connect this more strongly back to user-centered efforts. Under the "Socio-Technical Processes Framework for Design" section, more context is needed about the framework used to address micro and macro ethics. While you mention the Sommerville and Dewsbury framework and its background, the paragraph ends a little abruptly: "This process guides decision-making." Please offer more clarity here on this final statement. What areas is your program looking to improve or address? Consider adding a section that reflects on this journey and where your future efforts are going toward. In your conclusion you state that the program is most satisfying for developers. It gives a few assumptions that may not exist across stakeholders -- be careful here.
Author Response
The Reviewer has made some extremely helpful comments about restructuring our manuscript. We have followed this advice. Specifically, we have:
Restructured the paper to make our approach clearer.
Described the factors leading to HabITec in more detail.
Described HabITec fully and provided an example to show how we use the frameworks.
We have expanded on the Hacker mindset and its relation to Universal design.
We have added further information to this case study about future decisions and directions for HabITec.